# Comparative Analysis of Soccer Performance Intensity of the Pre–Post-Lockdown COVID-19 in LaLiga™

**DOI:** 10.3390/ijerph18073685

**Published:** 2021-04-01

**Authors:** Abraham García-Aliaga, Moisés Marquina, Antonio Cordón-Carmona, Manuel Sillero-Quintana, Alfonso de la Rubia, Silvestre Jos Vielcazat, Fabio Nevado Garrosa, Ignacio Refoyo Román

**Affiliations:** 1Facultad de Ciencias de la Actividad Física y del Deporte (INEF-Sports Department), Universidad Politécnica de Madrid, 28040 Madrid, Spain; abraham.garciaa@upm.es (A.G.-A.); antoupm@gmail.com (A.C.-C.); manuel.sillero@upm.es (M.S.-Q.); alfonso.delarubia@upm.es (A.d.l.R.); ignacio.refoyo@upm.es (I.R.R.); 2Department of Competitions and Mediacoach, LaLiga, 28043 Madrid, Spain; sjos@laliga.es (S.J.V.); fnevado@laliga.es (F.N.G.)

**Keywords:** performance, elite soccer, match running, activity profile, physical demands

## Abstract

Severe acute respiratory syndrome coronavirus 2 (SARS-CoV-2) forced a stoppage in the 2019/2020 season of LaLiga™, possibly influencing performance indicators in the return to competition. Therefore, here, we evaluated whether the stoppage due to the coronavirus 2019 disease (COVID-19) lockdown influenced physical performance compared to the start of LaLiga^TM^ in terms of high-intensity efforts. Using a semi-automatic, multiple-camera system, running activities during 22 matches were analyzed. We compared the first 11 matches of the season (pre-lockdown) with the 11 matches just after the restart of LaLiga™ (post-lockdown). The results showed higher (*p* < 0.05) performance in the pre-lockdown period compared with the post-lockdown period, including in medium-speed running (14.1–21 km/h), high-speed running (21.1–24 km/h), and sprinting speed running distances (>24 km/h). However, the number of accelerations/min and decelerations/min were significantly higher during the post-lockdown period. Therefore, we conclude that the stoppage due to the COVID-19 lockdown generated lower physical performance in the post-lockdown period compared with the pre-lockdown period, most likely due to the accumulation of matches (congested schedules).

## 1. Introduction

Coronavirus disease 2019 (COVID-19), caused by severe acute respiratory syndrome coronavirus 2 (SARS-CoV-2), has created a global emergency, requiring many countries to impose a state of quarantine and confinement in which freedom of movement is restricted. These exceptional measures were taken by most health authorities in European countries, especially after the World Health Organization characterized COVID-19 as a pandemic on 11 March 2020 [1]. In Spain, a state of alarm was declared on 15 March, with a peak in infections in mid-April [2]. The last official match of the Spanish Professional Soccer League (LaLiga™) was played on 8 March, and matches did not resume until 11 June.

During this period, soccer players were unable to carry out their professional work in the usual way, leading to different training protocols (i.e., training with no collaboration–opposition) that are completely different from competition [3,4,5]. 

Soccer is a complex sport of interaction between players (collaboration–opposition), in which players make a random transition between maximum, or almost maximum, multidirectional high-intensity effort and longer periods of low-intensity activity [6]. Players usually cover between 9 and 14 km in total during a match, with high-intensity runs that represent approximately 10% of this distance [7,8]. The amount of high-intensity exercise performed by players has been identified as the activity that exceeds a certain speed threshold [9,10].

These intermittent actions can be divided into different intervals using speed thresholds: medium-speed running (MSR) distance (14.1–21 km/h); high-speed running (HSR) distance (21.1–24 km/h); and sprinting-speed running (SSR) [11]. The distances covered at higher speeds are progressively smaller as the speed ranges increase [12]: 1614 ± 320 m for high-intensity runs [13]; 847 ± 349 m for very high-intensity runs [14]; and 184 ± 87 m for sprints [15]. Mallo, Mena, Nevado, and Paredes [16] suggest that the range of maximum speeds and the distance covered at very high intensity are crucial. Therefore, variables such as the number of sprints during a game and the maximum speed reached by an outfield player are very important when analyzing match performance data.

In Spain, on 11 May 2020, isolation measures were relaxed and professional soccer players competing in LaLiga™ were allowed to attend their clubs’ facilities. From this date, players were allowed to train, first using soccer-specific training routines with a “social” distance of 1 to 2 m (for 1 week), and then using training groups of an increasing number of players. Due to the positive evolution of the pandemic in Spain, and after adopting a strict protocol to minimize infection during soccer matches, the soccer and health authorities authorized the restart of LaLiga™ on 8 June 2020 to play the remaining 11 match days to complete the competition. Therefore, professional soccer players competing in LaLiga™ were confined to their homes for 8 weeks and allowed to train in preparation for the first match of the competition for 4 full weeks, for the total suspension of the competition, which lasted 12 weeks.

Many hypotheses have been proposed about the possible effect of the lockdown on the physical performance of players once the competition is restarted [3,4,5,10,17]. However, considering that the end of LaLiga™ usually takes place in the third week of May, and the beginning of the next season starts in the third week of August, there was a 12-week suspension from official league competition. Furthermore, in other cases, the transitional period between competitions is shorter due to participation in other official club tournaments, such as the Champions League, Europa League, European Super Cup, and Spanish Super Cup, or some official tournaments with national teams, such as the World and European Soccer Championships. It is important to note that, by agreement with the Real Spanish Soccer Federation (RFEF), the players must have at least one month (4 weeks) of vacation [18]. 

The pre-season period prepares players for the demands of the upcoming season. Comprehensive training that is focused on developing the conditions of the sport can help elite soccer teams to stay healthier during the season, so more sessions reduce the risk of injury [19]. In professional sport, the pre-season is a brief period when there is an increase in training load and intensity compared to the typical weeks of the season [20,21]. The aim of this phase is to stress the body and facilitate positive adaptations in the players’ fitness, which then have to be maintained during the season [22,23].

Due to the exceptional situation experienced during the season, which has been interrupted, and given the need to carry out a preparatory period to be able to confront the 11 matches remaining before the end of the league championship, it is considered necessary to analyze the physical performance of the Spanish LaLiga™ matches after both preparatory periods: pre-season and post-COVID-19 lockdown. The aim of this study was to evaluate whether the stoppage due to the COVID-19 lockdown influenced the players’ physical performance compared to the start of LaLiga^TM^ in terms of high-intensity efforts. It has been hypothesized that, despite the similitude in terms of stoppage time, as the training conditions were different, the physical performance variables of the players after the COVID-19 lockdown were reduced compared to the analysis of the matches pre-lockdown.

## 2. Materials and Methods 

### 2.1. Experimental Approach to the Problem

A descriptive analysis of the physical activities performed by professional soccer players was carried out using physical performance data from the 20 LaLiga^TM^ teams. In this study, two phases or periods of LaLiga^TM^ 2019/2020 were analyzed: the first third period (pre-lockdown) from the 1st to the 11th match day (matches played in 11 weeks), and the last third period (post-lockdown) from the 28th to the 38th (matches played over 5 weeks). Eleven matches were chosen from each phase, because the return to the competition after the COVID-19 (SARS-CoV-2) lockdown took place on the 28th match day and the same number of matches per phase was established. This choice of matches was made to establish the same number of matches at the beginning of the season and after the COVID-19 lockdown. The choice of matches rather than total days was prioritized because the fixture schedule at the start of the season is less congested, and if the same number of days had been taken, the pre-lockdown phase would have had fewer matches because they took place every week, and in the COVID-19 lockdown phase, they took place twice a week.

### 2.2. Data Collection and Analysis 

Data were obtained from LaLiga, which authorised the use of the variables included in this investigation. Data were extracted by a valid and reliable multicamera tracking system and associated software (Mediacoach^®^, Madrid, Spain). Mediacoach^®^ records the position of each player on the pitch at 25 Hz using a stereo multi-camera system composed of two multi-camera units placed on either side of the midfield line. Each multi-camera unit contains three cameras with a resolution of 1920 × 1080 pixels, which are synchronised to provide a stitched panoramic picture [24]. The analysis included physical performance data that were evaluated by a computerised multi-camera tracking system (TRACAB^®^, Stockholm, Sweden) that had a sampling frequency of 25 Hz. This tracking system semi-automatically evaluated the match performance data of all players, the position of the ball, and the corresponding match events, allowing the intensity of the run to be quantified. The validity of Mediacoach^®^ to assess running distance during soccer match play has been obtained through a high agreement with data obtained using global positioning system units [25,26] and data obtained from a reference camera system (i.e., VICON motion capture system [27]).

Running performance was assessed following previous research [11]. The variables analyzed were as follows: duration (min) is the sum of running speed analyzed per player and match played per team (min/11^2^; 11 matches × 11 players); the distance (km) is the result of the sum of the players of the team, showing the average of the teams per game in the period of time analyzed; distance (m/min) is the distance covered per minute of the sum of the 11 players per team of the 11 matches per team at the analyzed speeds; and medium-speed running (MSR) distance (14.1–21 km/h), high-speed running (HSR) distance (21.1–24 km/h), sprinting-speed running (SSR) distance (>24 km/h), high-intensity actions (>2.5 m/s^2^ and <−2.5 m/s^2^), accelerations (>3 m/s^2^), decelerations <−3 m/s^2^), maximum speed, and average speed (AVG speed) were also evaluated. Figure 1 shows the general experimental design of this study. 

### 2.3. Data Analysis

Data analysis was performed with IBM SPSS version 25.0 for Windows (IBM Corporation, Armonk, NY, USA). The assumption of normality was tested by Kolmogorov–Smirnov. For the comparison of the pre-lockdown and post-lockdown periods, Student t-tests for related samples were applied and the effect size was calculated using Cohen’s d. The effect size calculated with Cohen’s d was interpreted as: trivial <0.2; small = 0.01; moderate = 0.6–1.2; large = 1.2–2.0; very large = 2.0–4.0; and extremely large ≥ 4.0 [28,29]. Descriptive data for the quantitative variables are presented as mean (M) and standard deviation (SD). The level of statistical significance was set at α = 0.05.

## 3. Results

The physical performance of the LaLiga^TM^ teams was better in the first part of the season (pre-lockdown) compared to the last third period (post-lockdown) after the COVID-19 lockdown. More specifically, we found significant differences in the running variables MSR/min (14.1–21 km/h); HSR/min (21.1–24 km/h); SSR (m) (>24 km/h); SSR (count); high-intensity actions (count) (>2.5 m/s^2^ and <−2.5 m/s^2^); duration at running speed; total running distance covered, and running distance covered/min, as well as in accelerations (count) (>3 m/s^2^); decelerations (count) (>−3 m/s^2^); maximum accelerations (m/s^2^); and maximum decelerations (m/s^2^) (Table 1).

However, the number of accelerations/min and decelerations/min was higher in the post-lockdown phase compared to the pre-lockdown phase.

No significant differences (*p* > 0.05) were found in MSR (m); SSR/min; high-intensity actions/min; maximum speed (km/h), and AVG speed (km/h).

The calculated effect size showed an extremely large (≥4.0) value for the distance, large (1.2–2.0) for the duration, and moderate (0.6–1.2) for the MSR/min, HSR/min, SSR (m), SSR (count), high-intensity actions (count), accelerations (count), decelerations (count), max accelerations (m/s^2^), and max decelerations (m/s^2^).

## 4. Discussion

The aim of this study was to compare two blocks of 11 matches corresponding to the beginning of the season (pre-lockdown) and the end of the season (post-lockdown), corresponding to the first division of professional soccer in the Spanish LaLiga^TM^ in the 2019/2020 season. The main findings were significant in terms of higher performance data in the pre-lockdown phase in medium-speed running (MSR) and high-speed running (HSR), maximum acceleration, and maximum deceleration. On the other hand, in the post-lockdown phase, more significant results were obtained in acceleration/min and deceleration/min, and, although not significant, important results were obtained for the variables HSR/min and high intensity actions/min.

A comparison was made in terms of the high-intensity physical demands required in elite soccer. The reason for this analysis was the similarity between the start of the season (after a transition period) and the last third of the 2019/2020 season after a period of confinement. In both cases, there was a preparatory or pre-season phase, which usually ranges from approximately 4 to 7 weeks. Comparing these periods’ results with those of other studies that analyzed the risk of injuries is very interesting [30]. These risks could be produced by the individualized training conditions that were used during the confinement period. Many of these training sessions are far removed from the reality of the soccer context, where interaction with partners and opponents is demanded. These training sessions took place individually, as most of the individuals were locked down in apartments, with no access to open areas. This fact would make the intensity lower than in normal pre-season conditions. It is possible that other factors might have influenced this perception of the decline in performance after the return to competition, such as the absence of an audience or the change in the rules in terms of the number of substitutions made during the match.

The uncertainty of the return to competition meant that the teams had to train without being clear about the competition dates, which made their training difficult. In addition, soccer players are used to playing in a specific environment, with crowds in the stadiums, and this may be a psychological factor that should be studied in the future.

The results of this study showed that during the pre-lockdown phase, most of the physical performance variables measured were better compared to the post-COVID-19 lockdown. Only two variables (accelerations/min and decelerations/min) were significantly improved after the post-lockdown phase. No significant differences were obtained in SSR/min and high-intensity actions/min between the two periods.

This decrease in the physical variables after the lockdown might be because, as the season progresses, the performance diminishes due to the calendar being too condensed (too many matches in a short period of time). In the case of the first 11 matches of the season and the last 11 (the matches analyzed), this can be observed in matches played in 11 weeks and 5 weeks, specifically. Even though coaches look for player rotation in the line-ups, there is always a significant number of players who accumulate more minutes and starts, diminishing their performance in terms of high-intensity variables [31]. The reason that the variables SSR/min, high-intensity actions/min, accelerations/min, and decelerations/min increased in the post-lockdown phase may be the change in the regulation that offers the opportunity to replace five players in three stoppage windows. 

The similarities in the length of the interruption of the competition due to COVID-19 in Spain and the summer vacations between soccer seasons indicate that the players face a similar scenario at the beginning of the soccer season. This was similar in terms of time, but not in the case of training, in which the confinement situation was quite far from standard. There were also differences during the ‘pre-season’ due to the protocols that had to be followed in the different LaLiga^TM^ teams. It has been hypothesised that when the competition was resumed after the COVID-19 lockdown, professional LaLiga^TM^ players experienced physical challenges similar to those they usually experience during the first official matches of the season. The results of this study differ from those found by Brito Souza, López-Del Campo, Blanco-Pita, Resta, and Del Coso [32], which showed that, in the previous four competitive seasons of LaLiga^TM^, the physical performance of the players was lower at the beginning of the season, and the teams needed approximately 8–10 match days to achieve constant performance. 

During the traditional season, the decline in training performance happens at the end of the league competition or due to an injury or illness. These “de-training” situations are common and are not comparable to the situation caused by the confinement experienced at the beginning of the COVID-19 pandemic, despite the home training conducted [33].

Teams are already established in terms of tactics and strategies, but if not trained, team coordination can be affected. An important aspect was that the training protocols established by health authorities for all were the same in terms of the training situations that could occur. In addition, the players—first individually, then as a group, and then collectively—were preparing for the competition. The level of training volume differed from other pre-seasons. However, the types of exercises were not related to one another when approaching or moving away from the specifics of the game. This influences not only physical fatigue but also psychological fatigue, generating much more stress and a feeling of intensity if training is done in isolation. A very important factor was that these collective situations could not be addressed in friendly matches, something that is done in the pre-season. Therefore, the physiological and tactical volume was affected. In this way, the type of training influences the demands encountered in competition. The return to training protocol designed by LaLiga under the recommendations/limitations of the health authorities affects the training and, therefore, the conditional performance during the competition. In this context of limited training sessions, in terms of the number of participants and the limited number of preparation matches, there has been an increase in the number of small-space tasks with few players and a decrease in the number of long-distance tasks. In this sense, tasks with few players and a relatively small space per player could be an indicator of a deficit in the distances covered at high speeds; it is logically hard to reach high speeds in a small space [34,35]. Moreover, training with small groups of players, usually through tasks with more space limitations, promotes better adaptation to the ability to accelerate and decelerate repeatedly, which could be another reason for an increase in accelerations and decelerations per minute [36,37,38].

Although it is often postulated that a high performance value in the running of matches is important to achieve victory in a soccer competition, it has been shown that running performance during matches is not associated with the qualifying position at the end of the league [39]. Conversely, the most successful soccer teams covered a greater distance with the possession of the ball—in particular, at ≥21 km/h—than the less successful teams, and the running distance with the ball correlated positively with the points at the end of the season [32,39]. However, running with and without possession of the ball is not the same [40].

There is evidence that 4 weeks of de-training in soccer players causes an increase in the sprint time of 20 and 30 m and in the level of performance in an agility test [41]. Taking into account these data and considering that a long period of de-training, such as that caused by the COVID-19 pandemic, may result in a decrease in the number of fast-twitch muscle fibers [42,43], it is reasonable to expect a noticeable loss of athletic speed skills. Several studies have shown a close relationship between maximum speed values and muscle power values [44,45,46].

Furthermore, the decrease shown in high-intensity physical variables may be due to the influence of competition schedules and, subsequently, increased match congestion (typically, weeks with two or more matches, with less than 72 h between matches), which affected these variables to a greater extent [15,47,48]. Moreover, the Spanish soccer authorities have included other measures to alleviate the physical strain induced by official matches, such as increasing the number of player substitutions to five per match [49,50], refreshment breaks during the game, and timetables to reduce the number of hours played at higher ambient temperatures, all to reduce physical strain during the game.

The return of the teams to the post-confinement competition could be expected to result in a decline in the intensity and pace of play. If we add to this the lack of an audience and the influence that this may have on the teams playing at home [51], the perception of the external viewer would be of a decrease in the conditional play–performance rhythm.

Determining the correspondence between perception and reality is a very interesting object of study, and comparing this with what happens in the same period of the post-summer season could help to better explain these results. By doing so, this might open future lines of research that could contribute to improving collective physical performance (e.g., adjusting training loads for optimization, better recovery strategies between matches, etc.). This will help coaches to find solutions regarding the use of players in each of the competition scenarios presented to soccer teams.

This study sheds light on the potential consequences derived from COVID-19 confinement and its implications for professional soccer players’ performance. Nonetheless, there are some limitations to be considered. Not only does the confinement affect the decontextualization of the physical intensity data outside the context of the match sporting performance, but also score, weather, expulsion of teammates or opponents, and the condition of the field must be considered significant factors within intensity variables. Moreover, situational variables such as the location of the match or the absence of an audience, as is the case during this pandemic situation, need to be taken into account. 

## 5. Conclusions

Indicators of physical performance in LaLiga^TM^ matches after confinement due to COVID-19 varies conditional response in competition due to limitations in training methods during the 'pre-season' without experiencing low or negative performance in competition. After confinement physical performance were significantly lower at medium and high intensities. However, this was not the case for the number of accelerations/min and decelerations/min, which increased significantly in the matches during the post-lockdown phase. Most likely, players did not experience the best conditions, not only during training confinement but also in the specific post-COVID-19 reintroduction to the collective training sessions. Consequently, coaches should focus on different strategies to maintain the intensity levels of their teams, such as using rotations in the starting line-ups and making suitable substitutions. All of these actions together could help to obtain better performance during the official competitions.

## Figures and Tables

**Figure 1 ijerph-18-03685-f001:**
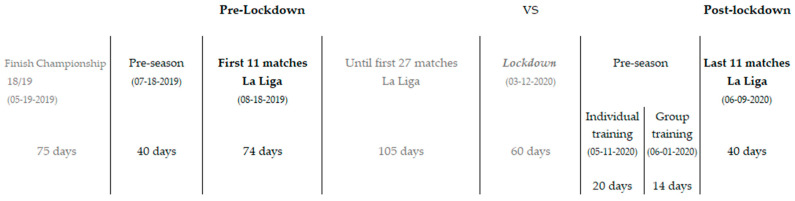
Timeline of the experimental design. The different periods have been included (pre-season, competition, lockdown, re-training, post-lockdown—resuming the competition). The different colours in the figure indicate the similarity between periods. In bold are highlighted the two periods that are compared. The following black degradation marks the two pre-season phases. Finally, the grey colour indicates the periods of the season where there is a break in the competition and the matches that have not been taken into account in the analysis.

**Table 1 ijerph-18-03685-t001:** Physical performance data. Descriptive statistics average ± standard deviation (M ± SD), *p*-value, and Cohen’s d.

Variables	Phases of Season					
	Pre-Lockdown		Post-Lockdown			
	M ± SD	*n*	M ± SD	*n*	*p*	Cohen’s d
Duration (min)	1852.96 ± 85.63	11	1712.49 ± 91.67	11	<0.001	1.58
Distance (km)	75.62 ± 0.58	11	70.20 ± 1.27	11	<0.001	5.5
Distance (m/min)	7699.22 ± 199.63	11	7037.74 ± 216.19	11	<0.001	3.18
MSR (m)	104.42 ± 2.12	11	103.59 ± 2.70	11	0.165	0.34
MSR/min	202.91 ± 19.80	11	186.67 ± 22.12	11	0.001	0.77
HSR/min	36.36 ± 2.40	11	33.58 ± 2.51	11	<0.001	1.13
SSR (m)	418.44 ± 28.33	11	382.65 ± 32.83	11	<0.001	1.17
SSR/min	6.06 ± 0.41	11	6.16 ± 0.51	11	0.308	0.27
SSR (count)	24.89 ± 1.1.48	11	23.06 ± 1.59	11	<0.001	1.19
High-Intensity Actions (count)	111.8 ± 6.06	11	105.17 ± 6.51	11	<0.001	1.05
High-Intensity Actions/min	1.48 ± 0.08	11	1.50 ± 0.09	11	0.139	0.23
Max. Speed (km/h)	29.91 ± 0.23	11	29.81 ± 0.36	11	0.115	0.33
Avg. Speed (km/h)	6.27 ± 0.13	11	6.22 ± 0.16	11	0.184	0.34
Accelerations (count)	47.07 ± 2.52	11	44.62 ± 2.53	11	<0.001	0.97
Decelerations (count)	52.76 ± 3.00	11	49.39 ± 3.47	11	<0.001	1.04
Accelerations/min	0.66 ± 0.03	11	0.68 ± 0.04	11	0.001	0.57
Decelerations/min	0.74 ± 0.04	11	0.75 ± 0.05	11	0.017	0.22
Max. Acceleration (m/s^2^)	5.65 ± 0.08	11	5.55 ± 0.10	11	0.007	1.10
Max. Deceleration (m/s^2^)	−6.04 ± 0.07	11	−5.98 ± 0.10	11	0.014	0.7

Duration: duration at running speed analyzed; distance: total running distance covered at running speed analyzed; MSR: medium-speed running; HSR: high-speed running; SSR: sprinting speed running.

## Data Availability

Not applicable.

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
