# Peer review of "Comparative Analysis of Soccer Performance Intensity of the Pre–Post-Lockdown COVID-19 in LaLiga™"

_ijerph, 2021, doi:10.3390/ijerph18073685_

Round 1

Reviewer 1 Report

I want to thank to the authors their effort to shed some light on one of the most challenging periods that the professionals had to face. Basically, the paper is, in my opinion well elaborated. Nevertheless, there are some points that need further discussion and some flaws that need to be addressed. I hope the following remarks can help the authors to improve their manuscript.

Abstract

This sentence is not clear, re-do: “The present study aimed to make a comparison of both periods of physical demands in terms of high-intensity efforts”

Introduction

“The state alarm” is a legal tool but the restrictions that affect the activity are far more important to state the question, which are the suspension of the league and the lockdown.

The restrictions in the first phase of the return to training although are considered, are a crucial difference to a regular pre-season and, in my opinion, should be highlighted and further discussed.

In the last paragraph of the introduction “This period of pre-season …”, although both periods can be considered as a preparatory phase, I think is much more interesting to point out the differences i.e. restrictions, duration, previous ‘off-season’ time, …

Methods

The authors choose to compare the initial 11 games of the season to match the number of games that were played in the post-COVID period, although I like the choice, I would like to read why they discard other options like:

  1. Simply both, before and after with different number of games
  2. Takin’ the same number of days instead of the same number of games

Results

Check how to report t-values and be consistent along the whole paper.

In table 1:

  • First of all, I would add the exact value of p-value in any comparison, you can set a minimum value i.e. p < 0.01 for the lowest values
  • Cohen’s d can also be included
  • Chose the right decimal positions i.e. I don’t really know if it is necessary to know how many centimeters have been recovered in a span of kilometers
  • Distance and distance·time-1 are wrong. Check
  • Follow the style guides when reporting numbers
  • Check the legend

Discussion

I missed in the discussion chapter a deeper comparison of the differences between a regular off-season/pre-season periods with the COVID-19 lockdown/return to play. One of the main factors that can be highlighted is that in the off-season players are allowed to do their normal activities in their daily life. Most of the individuals were locked down in apartments with no access to open areas i.e. garden or balcony. This difference can account for a lot of daily activity leading to more pronounced loss of performance in the same amount of time.

The uncertainty has been also a constraint during the return-to-train. When the athletes were asked to start training again, the competition plan and calendar were not clear yet, making the training plan more difficult to design.

Just a suggestion. The authors state that “teams needed approximately 8-10 match days to achieve a constant performance”. Since they have all the data a time-series analysis or just plotting how data evolves through time could have been added value.

One of the main changes of the league has been more congested fixture to finish the competition calendar. There are teams that are used to compete twice a week (Champions and Europa leagues), does this provide further advantage?

During the return-to-train period after the lockdown, they were many differences, one that favors this period is that the team is already build up in terms of tactics and strategies. Discuss. A second one was the limitation of time due to the limitation to one or small groups of people during the firsts weeks of preparation. This can have impact in the total volume in my opinion. I would like to read the authors’ discussion.

I would like to read, in opinion of the authors, which could be the effects of the change of substitutions during the game. They only point out that the change has been made but, if important, this can carry some consequences, and even lead to different strategies that could be a source of variation between periods.

Reviewer 2 Report

General Comments:

The current paper examined the differences in physical demands in terms of high-intensity efforts between pre- and post-COVID-19 lockdown. The paper is well written, but has a few editorial issues (see Specific Comments). Probably the biggest issue is that the authors cite a paper that they state analyzed injury risk, when the paper in question did not have any data or perform any analyses. As stated in the Specific Comments the cited paper was merely a position paper. A paper cited within the reference in question did do an analysis similar to what the authors suggest and perhaps they may wish to cite it instead.

Specific Comments:

Abstract Line 2: Suggest "of" in place of "at"

Abstract Line 4: Suggest "running" in place of "race"

Intro Paragraph 2 Line 1: Suggest "were unable" in place of "have not been able".

Intro Paragraph 3 Line 4-5: Suggest "with high-intensity runs that represent"

Results Paragraph 1 Line 1: "TM" is not formatted in other instances.

Results Paragraph 4 Line 1: There appears to be a typo ".80" for the effect size, as defined previously.

Table 1 footnotes: Why are there two ways of indicating "p<.001"? Is there a typo? Also, "p<0.05" includes a zero before the decimal, while the others do not; please be consistent.

Discussion Paragraph 2 Lines 5-6: I don't think it correct that the authors use this citation to imply that the paper by Casais-martinez and Lago-Peñas (21)  analyzed risk of injury. The paper was merely a position paper and did not perform any analyses to data that they presented or to any other papers. If the authors wished to use the paper cited by Casais-martinez and Lago-Peñas to make that case, that would be appropriate and is:  Myer GD, Faigenbaum AD, Cherny CE, et al. Did the NFL Lockout expose the Achilles heel of competitive sports? J Orthop Sports Phys Ther. 2011; 41:702-5.

Discussion Paragraph 8 Line 1: Suggest "de-training" in place of "untraining".

Reviewer 3 Report

The topic is interesting. However the authors assessed players performance by video analysis and it may differ from experimental and individual analysis such as GPS use. 

Specific comments below:

Introduction paragraph 7. what were the hypothesis? Please describe and speculate.

It no clear the research gap. The authors want to understand the effects of stopping La Liga or compare the results with the vacancies? The authors may specify and clarify the research gap.

Methods: Is a 2% error significant in elite sports?

The data collection must be more detailed. In fact the authors link to reference 11 and 19. However the methods lacks of information.

This reviwer suggest to use different effect sizes qualification cutt of values based on elite sports.

The results must be presented with descriptive analysis and then comparisons. It would be better to present more visual information (such as graphics). The results must be improved.

Discussion: 2nd paragraph should be about how the authors assessed the results? Is the method the gold standard? If not how can results be affected? Is possible to control errors? What literature says?

This reviwer did not find the limitations of this study.

Round 2

Reviewer 3 Report

The authors have assessed my qualifications. However, in conclusion, please specify if the confinement lead to pre-season baseline performance. 
